# Human Papillomavirus-Associated Oropharyngeal Squamous Cell Carcinoma: A Systematic Review and Meta-Analysis of Clinical Trial Demographics

**DOI:** 10.3390/cancers14164061

**Published:** 2022-08-22

**Authors:** Tamar M. Gordis, Joshua L. Cagle, Shaun A. Nguyen, Jason G. Newman

**Affiliations:** Department of Otolaryngology—Head and Neck Surgery, Medical University of South Carolina, Charleston, SC 29425, USA

**Keywords:** clinical trial, oropharyngeal cancer, HPV-associated cancer

## Abstract

**Simple Summary:**

HPV-associated oropharyngeal squamous cell carcinoma (OPSCC) is unique amongst oropharyngeal cancers in its high responsiveness to treatment and its lower mortality rate. As a result, numerous clinical trials have been conducted to identify treatment modalities and protocols. In order for these trials to have meaningful impact on HPV-associated OPSCC patients, proper demographic representation by trial participants is essential. The aim of our systematic review and meta-analysis was to assess the demographics of trial participants for HPV-associated OPSCC clinical trials and compare them with those reported by national databases. We determined that clinical-trial participants were predominately non-smoking white men, with tonsils as the primary tumor site. These findings reflect the demographics reported by the National Cancer Database. Our results imply that HPV-associated OPSCC clinical trials appropriately represent the target population and offer immense benefit.

**Abstract:**

The objective of our paper was to answer the following question: how do patients with HPV-related oropharyngeal squamous cell carcinoma OPSCC (Population) enrolled in clinical trials (Intervention), compared with national database reports of HPV-associated OPSCC patients (Comparison), present demographically (Outcome)? We conducted a systematic review and meta-analysis of studies pertaining to clinical trials of HPV-associated OPSCC and participant demographics in the United States. PubMed, Scopus, CINAHL, and the Cochrane Library were searched from inception to 2 February 2022. Studies of overlapping participant cohorts and/or studies conducted outside of the United States were excluded. Primary outcomes were patient age, sex, and race. Secondary outcomes were smoking history, alcohol history, history of prior cancer, and tumor origin site. Meta-analysis of single means (mean, N for each study, and standard deviation) for age, pack years, and smoking years was performed. Pooled prevalence rates of gender, race, alcohol history, tobacco history, and tumor origin site were expressed as a percentage, with 95% confidence intervals. Meta-analysis found patients to be predominately non-smoking white males, with tumors originating from the tonsil. Our findings reflected the demographics reported by the National Cancer Database (NCDB) for HPV-associated OPSCC. This indicates that HPV-associated OPSCC patients are appropriately represented in clinical trial demographics.

## 1. Introduction

In the last two decades, there has been an exponential rise of oropharyngeal squamous cell carcinoma (OPSCC), despite simultaneous decreases in both head and neck cancer mortality rates and rates of cigarette use in the United States [1,2]. This is largely due to elevated rates of oropharyngeal infection with oncogenic HPV strains. Regardless of origin, OPSCC poses notable potential for mortality, along with dramatic impairments in one’s daily functions and abilities [3]. Furthermore, treatment options, ranging from surgery to intensity-modulated radiotherapy and concurrent cisplatin, carry significant burden; surgeries include the risk of postoperative complications, potential rehospitalization, and significant postoperative disability, while chemo-radiotherapy (CRT) can result in treatment-induced toxicities, such as mucositis, dysphagia, xerostomia, and dysgeusia. 

Amongst OPSCC’s variety of etiologies, human papilloma virus (HPV) holds particular interest and promise. HPV-associated OPSCC, especially of the p16/HPV-DNA subtype, demonstrates increased cellular chemo- and radio-sensitivity, corresponding to higher response rates and greater reductions in both disease progression and fatality. HPV-related OPSCC’s ubiquitous responsivity to surgery (especially compared to non-HPV-associated OPSCC), combined with the morbidities of current guidelines, have catalyzed numerous clinical trials in efforts to identify opportunities for CRT de-intensification and less-invasive surgeries. Recent clinical trials, including ORATOR, MC1273, and AVOID, have focused on HPV-related OPSCC management and have compared outcomes of chemoradiation with transoral robotic surgery and deintensification of adjuvant therapies, with progressively more trials in recruitment [4]. Although these trials have demonstrated significant promise, questions about population representation from trial to clinic remain. To best utilize clinical trial findings in treatment protocols for various patients, it is essential that the trial demographics are representative of said patients.

Instances of discrepancies between trial participants and patient populations have been well-documented throughout multiple specialties. Heiat et al. compared the demographics of patients in heart-failure-related randomized control trials with those of the general population and found that trial participants markedly differed from the general population, with an overrepresentation of white and male patients [5]. Johnston et al.’s systematic review of sex, age, race, and intervention type in clinical trials for HIV determined that females, older patients, and non-white patients were underrepresented in trial populations [6]. In reviewing randomized clinical trials for lipid-lowering therapies, Khan et al. noted consistent underrepresentation of female and older patients, limiting the evidence base for efficacy and safety in the treatment of these patient group [7]. Conversely, Strait et al.’s systematic review of participant demographics for rheumatoid arthritis randomized clinical trials found that males and nonwhite patients were significantly underrepresented in comparison to national statistics [8].

Currently, there are no equivalent reviews of demographics in clinical trials for HPV-associated OPSCC. The goal of this study was to ascertain the demographics of participants in HPV-OPSCC clinical trials in the United States and compare with those of national databases.

## 2. Materials and Methods

### 2.1. Systematic Literature Search

This systematic review/meta-analysis aimed to answer the following question: how do patients with HPV-related OPSCC (Population) enrolled in clinical trials, (Intervention), compared with national database reports of HPV-related OPSCC patients (Comparison), present demographically (Outcome)? A detailed search strategy (Appendix A) was developed in the following four databases: PubMed (National Library of Medicine, National Institutes of Health), Scopus (Elsevier), CINAHL (EBSCO), and Cochrane Library (Wiley). The search strategy used a combination of subject headings (e.g., Medical Subject Headings [Mesh] in PubMed). The PubMed search strategy was modified for the other three databases, replacing Mesh terms with appropriate subject headings, when available, and maintaining similar keywords. The databases were searched from inception through 7 February 2022, and results were limited to English language and clinical trials. References were uploaded to Covidence systematic review software (Veritas Health Innovation, Melbourne, Australia) and screened for relevance.

### 2.2. Selection Criteria

Abstracts were first independently reviewed by two reviewers (T.M.G. and R.G.) to identify all studies pertaining to clinical trials of HPV-related OPSCC and participant demographics. Non-English studies, review articles, nonhuman studies, non-journal articles (e.g., abstract only), and studies of clinical trials conducted outside of the United States were excluded. Studies that included overlapping participant cohorts with other trials were also excluded. Any conflicts were resolved by discussion. To identify additional articles, the reference lists of relevant articles were hand searched. 

### 2.3. Data Collection

Data included in the analysis and discussion were extracted by two reviewers (T.M.G. and R.G.). Disagreements were resolved by discussion. Primary outcomes were patient age, sex, and race. Secondary outcomes included smoking history, alcohol history, history of prior cancer, and origin site of HPV-related OPSCC, classified as tonsil, base of tongue (BOT), both sites, or not otherwise specified (NOS). In instances of incomplete data, an attempt was made to contact the primary author via email for clarification or sharing of primary data.

### 2.4. Statistical Analysis

Meta-analysis of single means (mean, N for each study, and standard deviation) for age, pack years, and smoking years was performed by Comprehensive Meta-Analysis version 3 (Biostat Inc., Englewood, NJ, USA). Meta-analysis of proportions was performed using MedCalc 19.6 (MedCalc Software Ltd., Ostend, Belgium; https://www.medcalc.org; accessed on 2020). The pooled prevalence rate of gender, race, alcohol history, tobacco history, and tumor origin site were expressed as a percentage with 95% confidence intervals (CI). Each measure was weighted according to the number of patients affected. The weighted-summary proportion was calculated by the Freeman–Tukey transformation [9]. Heterogeneity among studies was assessed using χ^2^ and I^2^ statistics. I^2^ < 50% indicated acceptable heterogeneity, and, therefore, the fixed-effects model was used. Otherwise, the random-effects model was performed. Finally, Egger’s tests with funnel plots were performed to further assess the risk of publication bias [10,11]. In a funnel plot, the treatment effect is plotted on the horizontal axis, and the standard error is plotted on the vertical axis. The vertical line represents the summary estimated derived using a fixed-effect meta-analysis. Two diagonal lines represent (pseudo) 95% confidence limits (effect ± 1.96 SE) around the summary effect for each standard error on the vertical axis. These show the expected distribution of studies in the absence of heterogeneity or selection bias. In the absence of heterogeneity, 95% of the studies should lie within the funnel defined by these diagonal lines. Potential publication bias was evaluated by visual inspection of the funnel plot, as bias results in asymmetry of the funnel plot, and Egger’s test, which statistically examines this asymmetry. A *p*-value of <0.05 was considered to indicate a statistically significant difference for all statistical tests.

## 3. Results

This study was conducted in accordance with the Preferred Reporting Items for Systematic Reviews and Meta-Analyses (PRSIMA) guidelines (Figure 1). A total of 32 studies with 2995 patients were included in the review [2,12,13,14,15,16,17,18,19,20,21,22,23,24,25,26,27,28,29,30,31,32,33,34,35,36,37,38,39,40]. Table 1 shows the year, location, trial identifier (NCT number, RTO, etc.) or study identifier, and trial phase for each included study.

Of the 2995 included patients, 2918 patients were reported by sex, 2457 were reported by age, 1993 were reported by race, 2668 were reported by tobacco and alcohol history, and 2162 were reported by primary tumor site. The mean patient age was 59.1 [51.9, 66.2] years (Figure 2) and the mean pack years for smoking history was 4.2 [−2.6, 11.1] years (Figure 3). Overall, patients were found to be significantly predominately white and male, with no history of smoking tobacco, a current drinking status, and tumors originating from the tonsil. A funnel plot with Egger’s test (−0.11, 95% CI −1.10 to 0.88, *p* = 0.82) demonstrated all studies were within the funnel, suggesting little publication bias (Figure 4). Table 2 shows the meta-analysis of sex and race, Table 3 shows the meta-analysis of tobacco and alcohol history, and Table 4 shows the meta-analysis of tumor origin site. I^2^ values indicated high levels of heterogeneity amongst patients in regard to racial classification as African American, Hispanic, and other/NOS, smoking and alcohol history, and tumor origin site. 

A review of the HPV-associated OPSCC population within the oropharyngeal cancer cohort in the National Cancer Database (NCDB) reported the demographics of 14,805 patients: patients were a mean age of 58.4 ± 9.5 years, 67.9% (*n* = 12,600) male, 67.7% (*n* = 13,479) white, and 69.7% (*n* = 8068) of high socioeconomic status, with 65.2% (*n* = 5598) having tumors originating from the tongue base [41,42].

## 4. Discussion

The objective of this study was to answer the following question: how do patients with HPV-related OPSCC (Population) enrolled in clinical trials (Intervention), compared with national database reports of HPV-related OPSCC patients (Comparison), present demographically (Outcome)? With the determination of HPV status for OPSCC rapidly becoming the standard of care, given HPV-positive patients’ approximate 60% reduction in risk of death and absolute survival difference of nearly 30% at five years, the importance of reviewing the demographics reflected in current advances for treatment cannot be understated [43]. Furthermore, even though overall recurrence rates are lower for HPV-positive OPSCC patients, they have a higher proportion of recurrences at distant sites than HPV-negative patients and are more likely to experience disseminated metastasis in non-traditional/non-pulmonary sites [44,45,46]. Thus, demographically-appropriate treatment modalities are essential.

According to Pytynia et al., the prototypical patient with HPV-positive OPSCC is a nonsmoker with a history of multiple sexual (oral and/or genital) partners [44]. Furthermore, associated tumors are most likely to originate in the tonsil or base of tongue [47]. This is reflected in the demographics of our included trials, as patients were predominantly Caucasian, male, and non-smokers, with tumors originating in nearly equal proportions from the tonsil or base of tongue. Our findings, therefore, suggest that HPV-associated OPSCC patients are, in fact, properly represented in clinical trials.

We were unable to perform analysis on sexual partner histories or socioeconomic status, as too few studies reported these variables. HPV-associated OPSCC differs drastically from other cancers in its incidence amongst Caucasians relative to non-Caucasians; whereas Camidge et al. reported that African American men are more likely to have malignant tumors and lower rates of survival than the general population, Pytynia et al. found that the stark preponderance of HPV-associated OPSCC in Caucasian males over the general population actually contributes to the diminishment in cancer-related disparities between Caucasian and African American males [44,48]. Although Kennedy-Martin et al. found that RCT participant populations are highly selective and have lower risk profiles than that of the general population, they did so in the context of cardiology, mental health, and general oncology [49]. Heiat et al. and Varma et al. demonstrated disparities in representation for clinical trials regarding cardiology and oncology, respectively, with trial demographics that were predominately Caucasian, despite significant incidence in African American and other non-Caucasian patients [5,50]. Conversely, HPV-associated OPSCC has been described with a more skewed incidence amongst Caucasian males.

While our findings indicate that the HPV-associated OPSCC population is adequately represented in clinical trials, it is also worth considering areas of potential underreporting in the general population, such that certain demographics are going unnoticed. Dunlop et al. and Dovido et al. showed that non-Caucasian patients have fewer physician contacts, utilize fewer hospital and outpatient surgery services, and are more likely to suffer from unequal geographic distribution of medical services [51,52]. Barriers that present before an initial clinical encounter could result in an underreporting of HPV-associated OPSCC cases in select patient groups, translating into underreporting of clinical trial participation and poor representation in treatment evaluations. 

The largest limitation of this study was the lack of a proper bias assessment tool. While this was a meta-analysis of clinical trials, models such as ROBINS-I or Cochrane Collaboration’s tool were not appropriate, as they utilize pre-intervention, peri-intervention, and post-intervention metrics [53]. Since we focused solely on the demographics of each trial and not the specific potential deviations from intended interventions or protocol, we were unable to utilize the standard risk of bias assessments. A uniform assessment method that allows for analysis without the inclusion of specific interventions would allow for risk of bias to be determined in this context.

Disparity in patient representation for clinical trials is, unfortunately, a relatively familiar dilemma; Clark et al. reported that African Americans and Hispanics comprised 13% and 16% of the US population in 2016, respectively, yet made up only 5% and 1% of clinical-trial participants, respectively. In reviewing the barriers faced by patients in clinical trial representation, Clark et al., along with Ford et al., found that the five primary obstacles were physician mistrust, discomfort with the clinical trial process, lack of information, time and resource constraints, and lack of awareness of resources [54,55]. A review of Phase I–III trials for drugs targeting breast, colorectal, lung, and prostate cancer by Ramamoorthy et al. found that 79.7% of the trial participants were Caucasian, 12.4% Asian, 3.8% African American, and 3.6% Hispanic [56]. Similarly, Loree et al.’s review of clinical trials for FDA approvals for cancer drugs reported that Caucasians represented 76.3% of the trial participants, followed by 18.3% Asians, 6.1% Hispanics, and 3.1% African Americans [57].

This study was also potentially limited by the heterogeneity in reporting demographics and histories. The majority of included studies reported sex, age, and tobacco use; however, race was less frequently reported and variables such as socioeconomic status and sexual partner history were even more rarely documented. Since the latter two variables have also demonstrated key patterns in the general population, the addition of more studies that further specify these characteristics would enable more detailed analysis of trial participant demographics, in comparison to the overall patient population. In addition to heterogeneity in frequency of reporting, there was also heterogeneity in the specifics of describing certain features. For example, some studies provided mean pack years, some reported quantities of patients with pack years greater than, equal to, or less than 10 years, and others reported the equivalent for 20 pack years. Since the data was presented differently, fewer analyses were done, as each sub-group had a smaller sample size. Future analyses would be improved by standardizing the format of reporting patient demographics and history.

## 5. Conclusions

Having been described as an epidemic, HPV-associated OPSCC is both increasingly common in incidence as well as promising in treatment responses, warranting various clinical trials for treatment evaluation [45]. The demographics of participants for said trials were reviewed and meta-analyzed to answer the following question: how do patients with HPV-related OPSCC (Population) enrolled in clinical trials (Intervention), compared with national database reports of HPV-related OPSCC patients (Comparison), present demographically (Outcome)? Overall, our findings revealed a predominance of middle-aged Caucasian males without a history of smoking, reflective of the demographics for HPV-associated OPSCC in the United States, as reported by the NCDB. Further studies could increase both the sample size and power of these trial demographics and allow for even better reflection of the general population in participant groups for clinical trials.

## Figures and Tables

**Figure 1 cancers-14-04061-f001:**
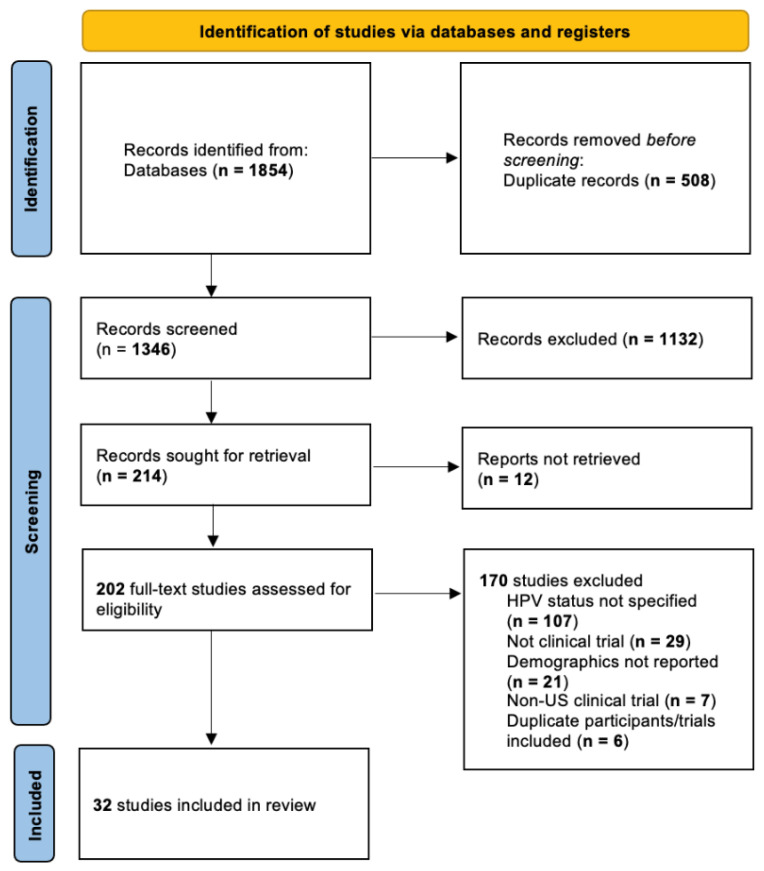
Preferred Reporting Items for Systematic Reviews and Meta-Analysis (PRISMA) flowchart for data search performed in PubMed (National Library of Medicine, National Institutes of Health), Scopus (Elsevier), CINAHL (EBSCO), and Cochrane Library (Wiley).

**Figure 2 cancers-14-04061-f002:**
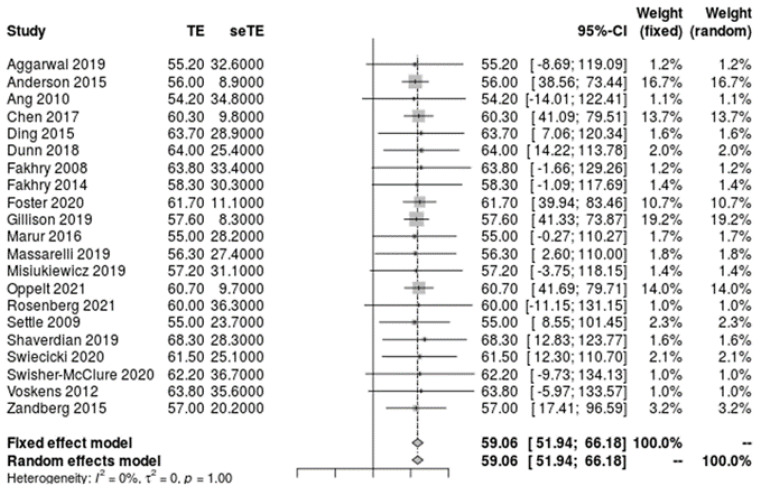
Meta-analysis of single means for participant age.

**Figure 3 cancers-14-04061-f003:**
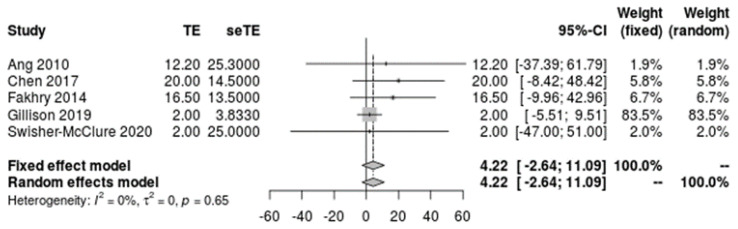
Meta-analysis of single means for participant pack years.

**Figure 4 cancers-14-04061-f004:**
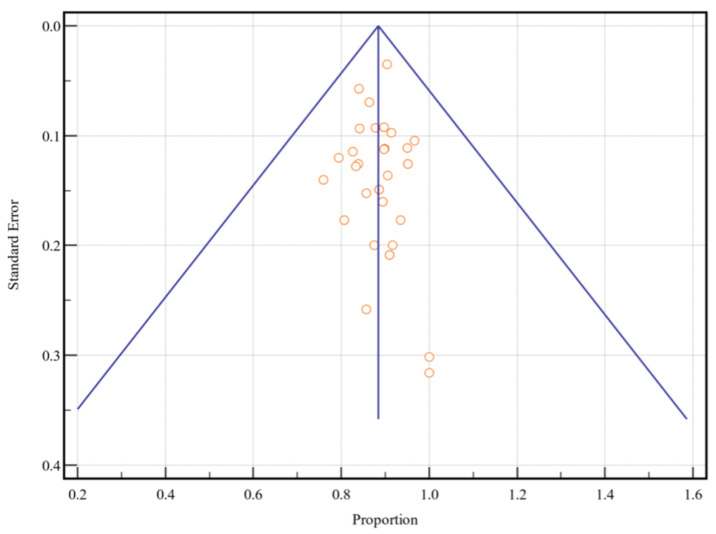
Funnel plot of all included studies.

**Table 1 cancers-14-04061-t001:** Clinical trial characteristics *.

*Author, Year*	State(s)	Trial ID	Phase(s)	Total *n*
*Aggarwal 2019*	PA	NCT02163057	I/IIa	22
*Anderson 2015*	MD, NY, OR	NCT01342978	NR	116
*Ang 2010*	TX	NCT00047008	III	206
*Chen 2017*	CA	NCT02048020/NCT01716195	II	44
*Chera 2015*	NC, FL	NCT01530997	II	44
*Chera 2019*	NC	NCT02281955	II	114
*Chera 2020*	NC, FL	NCT03077243	II	115
*Ding 2015*	TX	NCT01893307	II/III	31
*Dunn 2018*	NY	NCT01721525	Ib	10
*Fakhry 2008*	MD	ECOG protocol 2399	II	38
*Fakhry 2014*	MD	RTOG0129/RTOG0522	III	105
*Foster 2020*	IL	NCT02258659	II	62
*Gillison 2019*	TX	NCT01302834	NR	805
*Kumar 2008*	MI	UMCC9921	NR	50
*Ma 2019*	MN, AZ, FL	NCT01932697	II	79
*Marur 2016*	CT	NCT01084083	II	80
*Massarelli 2019*	TX	NCT02426892	II	22
*Miles 2021*	NY	NCT02072148	II	54
*Misiukiewicz 2019*	NY	NCT01706939	III	23
*Mowery 2020*	NC	NCT01908504	NR	62
*Oppelt 2021*	MO	NCT02101034	II	24
*Rosenberg 2021*	IL	NCT02258659	II	90
*Rosenthal 2016*	TX	NCT00004227	III	75
*Samuels 2016*	MI	PO1CA59827	II	53
*Seiwert 2019*	IL	NCT01816984	II	62
*Settle 2009*	MD	TAX 324	III	68
*Shaverdian 2019*	CA	NCT01716195	II	24
*Spector 2012*	MI	UCMCC02-021	II	78
*Swiecicki 2020*	MI	NCT01663259/NCT00904345	II	42
*Swisher-McClure 2020*	PA	NCT02159703	II	60
*Voskens 2012*	MD	NCT00257738	I	31
*Yom 2021*	CA	NCT02254278	II	306

NR = Not reported. * All trials were classified as Level 1 according to the OCEBM LOE.

**Table 2 cancers-14-04061-t002:** Meta-analyses of patient sex and race and I^2^ values amongst all studies and patients.

Identifier	Proportion % [95% CI]	I^2^ (%)
Male	88.2 [86.4, 89.9]	41.7
Female	11.8 [10.1, 13.6]	41.7
White	91.1 [88.9, 93.0]	44.1
African American	4.8 [2.7, 7.3]	76.9
Hispanic	1.8 [0.7, 3.3]	70.5
Asian	0.3 [0.1, 0.6]	0.0
Other/NOS	2.5 [1.4, 3.9]	58.7

**Table 3 cancers-14-04061-t003:** Meta-analysis of tobacco and alcohol history and I^2^ values amongst 28 studies (*n* = 2691).

History	Proportion % [95% CI]	I^2^ (%)
Current smoker	13.0 [5.9, 22.2]	87.8
Former smoker	30.6 [18.7, 44.0]	90.6
Never smoker	50.0 [43.0, 57.1]	90.2
Unknown smoking history	6.5 [3.8, 9.9]	0.0
History of <20 pack years	16.5 [6.2, 30.4]	83.8
History of ≥20 pack years	22.6 [12.7, 34.3]	80.3
History of ≤10 pack years	24.4 [17.6, 31.8]	87.1
History of >10 pack years	26.9 [19.2, 35.4]	87.8
Current drinker	46.2 [26.3, 66.8]	89.2
Former drinker	10.1 [1.5, 48.5]	95.2
Never drinker	28.7 [6.6, 58.4]	95.4

**Table 4 cancers-14-04061-t004:** Meta-analysis of tumor origin site and I^2^ values amongst 21 studies (*n* = 2162).

Tumor Origin Site	Proportion % [95% CI]	I^2^ (%)
Tonsil	44.4 [39.2, 49.7]	79.3
Base of tongue	41.4 [34.5, 48.5]	89.0
Tonsil and base of tongue	2.6 [0.5, 6.3]	93.4
Other/not otherwise specified	4.3 [2.3, 6.8]	81.6

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
