# Peer review of "Human Papillomavirus-Associated Oropharyngeal Squamous Cell Carcinoma: A Systematic Review and Meta-Analysis of Clinical Trial Demographics"

_cancers, 2022, doi:10.3390/cancers14164061_

Round 1

Reviewer 1 Report

OSCC-related to HPV infection is very significant problem. However, The authors did not write what is new in their study.
Whether the results of their analysis may have any significance in clinical practice
Are there any limitations to this paper?
After the above doubts are corrected, the article may be accepted for publication

Reviewer 2 Report

The topic is interesting and rarely thought of, however, I wonder what would be the impact out of answering this question.

Introduction: Suggest clarifying the clinical application of the findings that will come out from this analysis and how this is needed and would have an impact on the study population care.

Methods: It is well described in sufficient details. (1) Recommend quality control assessment to be added and applied. (2) Extra space in line 113. (3) Describe the formula used if raw data is not presented as mean and standard deviation. (4) There is redundancy and unneeded repetition form lines 121 to 130. (5) Meta-regression analysis is recommended.

Results: (1) Fig.1 list databases used in the workflow. (2) Table 2 and 3 identify the number of studies and number of study subjects used for the analysis of each parameter. (3) PICO stated in lines 82-84 does not match the output of one arm meta-analysis. The research question compared between two arms, so where is the arm of national databases. Only present description of one arm which is the clinical trials. Based on the question stated, it is needed to compare e.g sex in clinical trials x national databases, ....etc. (4) Data on heterogeneity was not supplied. (5) Meta-regression is recommended in the results. (6) Trial sequential analysis is also suggested.

Discussion: some sections represent ew values and should be shifted to the results.

Thank you

Round 2

Reviewer 2 Report

Request the table of input of smoking variable (pack year) used for Table 3.